# Higher-Order Patterns Reveal Causal Temporal Scales in Time Series Network Data

**Luka V. Petrović**
University of Zurich,
Zürich, Switzerland
petrovic@ifi.uzh.ch

**Anatol Wegner** and **Ingo Scholtes**
Julius-Maximilians-Universität Würzburg,
Würzburg, Germany
{anatol.wegner, ingo.scholtes}@uni-wuerzburg.de

## Abstract

The analysis of temporal networks heavily depends on the analysis of time-respecting paths. However, before being able to model and analyze the time-respecting paths, we have to infer the timescales at which the temporal edges influence each other. In this work we introduce an information theoretic measure, the causal path entropy, with the aim to detect the timescales at which the causal influences occur in temporal networks. The measure can be used on temporal networks as a whole, or separately for each node. We find that the causal path entropy has a non-trivial dependency on the causal timescales of synthetic and empirical temporal networks. Furthermore, we notice in both synthetic and empirical data that the entropy tends to decrease at timescales that correspond to the causal paths. Our results imply that timescales relevant for the dynamics of complex systems can be detected in the temporal networks themselves, by measuring higher-order correlations. This is crucial for the analysis of temporal networks where inherent timescales are unavailable and hard to measure.

The research of dynamic complex systems has in recent years advanced beyond static graphs [1, 2]. The focus has shifted to various generalizations of diadic interactions in graphs: multiple types of interactions in multilayer network [3], multibody interactions in the form of simplicial complexes and hypergraphs [4] and models that incorporate concepts of memory [5–7]. Such generalized relationships allow us to model richer data, without losing possibly important features of the data.

Temporal networks record not only who interacted with whom, but also when each interaction happened, which allows (and often requires) analysis beyond the standard network approach [8, 9]. The time information can yield valuable insights on its own [10], and, although the temporal and topological aspects of temporal networks were initially mostly studied independently, even richer insights are hidden in the coupling of the temporal and topological patterns. Such coupling can affect the statistics of time-respecting paths [8] in temporal networks and thus complicate the analysis of temporal networks, e.g., analysis of accessibility [11], reachability [12], spreading [5, 6, 13, 14], clustering [15], centralities [16], and visualization [17]. In cases when the statistics of time-respecting paths deviate significantly from the statistics of random walks in static graphs, the static graph can become a misleading representation of the temporal network.

Although, there are many possible ways in which temporal and topological patterns can couple in complex systems, one of the most basic cases is when the occurrence of a temporal edge causes a change in the frequencies of subsequent edges emanating from the target node within a given time-window. For instance, in a communication network we expect an incoming message to induce a outgoing message on the same topic, e.g. in the form of a reply, within a certain time window reflecting the minimal reaction time and memory of the recipient. Knowing the timescale at which such causal influence take place would allow us to capture the time-respecting paths that correspond to casual influences; this would in turn improve the analysis of the temporal network, e.g. the detection of time central nodes or community detection. However, information on the timescales relevant for the temporal network dynamics is rarely available in a real world settings.

Petrović et al., Higher-Order Patterns Reveal Causal Temporal Scales in Time Series Network Data (Extended Abstract). Presented at the First Learning on Graphs Conference (LoG 2022), Virtual Event, December 9–12, 2022.

We define an information theoretic measure aimed at detecting the prevalence of causal interactions at various timescales of complex systems. We demonstrate that our measure can be used to infer timescales that are relevant to the dynamics of temporal networks in both synthetic and real world data.

Let $\Gamma = (V, \mathcal{E})$ be a temporal network consisting of a set of nodes $V$ and a set of time-stamped edges $\mathcal{E} \subseteq V \times V \times \mathbb{R}$. A temporal edge $(v, w, t) \in \mathcal{E}$ represents a direct link from node $v$ to node $w$ at time $t$. For simplicity, we assume that the temporal edges are instantaneous, however the method and algorithms can be modified in a straightforward fashion to the case where edges have finite duration. Formally, we call a sequence of time-stamped edges $(v_1, w_1, t_1), \ldots, (v_k, w_k, t_k)$ a time-respecting path iff for all $i \in \{2, \ldots, k\}$ they satisfy the following conditions [8, 18, 19]:

$$w_{i-1} = v_i \text{ and } \delta_{\min} < t_i - t_{i-1} < \delta_{\max}. \tag{1}$$

The parameters $\delta_{\min}$ and $\delta_{\max}$ naturally introduce a timescale that affects all analyses of temporal networks that are based on time-respecting paths. Examples of such analyses include detection of cluster structures in temporal networks, measures of temporal centrality used to rank nodes in temporal networks, as well as results about dynamical processes like epidemic spreading, or diffusion processes. The timescale has to be defined differently for processes *on* the temporal network or the processes *of* the temporal network [8]. In the former case, the timescale is defined by the process running on the temporal network, e.g. in the case of an epidemic that is spreading over a temporal network of contacts, the timescale is a property of a disease, related to the time interval in which a person is contagious and not related to the timescales at which contacts occur [1]. In the latter case, the timescale is part of the process of edge activation, and thus shapes the temporal network itself, e.g. information that is spreading between persons is also affecting the persons' choice with whom to share the information: a person would be more likely to share the family-related information with a family member and work-related information with a colleague. We are investigating this latter case, more specifically, we consider the problem of detecting the time window $\Delta t = [\delta_{\min}, \delta_{\max}]$ at which causal correlations between temporal edges take place.

**Timescales in temporal networks.** In the literature, there exist a variety of definitions of timescales in temporal networks, as well as a variety of methods aimed at detecting them. The various definitions of timescales are based on the different structural features of temporal networks. One popular definition of timescales in temporal networks is the approach based on splitting the network into time-slices and aggregating the edges inside the time-interval [21]. In the same framework, Ghasemian et al. [22] and Taylor et al. [23] investigate the limitations of detectability of cluster structures dependent on the timescales of aggregation. Since this framework is based on aggregating the temporal network into a sequence of static time-aggregated networks, it loses information of the time-respecting paths and is therefore not in line with our aims. Other lines of research often related to timescale detection are change point detection [24], and analysis of large-scale structures. Gauvin et al. [25] detects clusters and their temporal activations in a temporal network using tensor decomposition. Similarly, Peixoto [26] proposed a method to detect the change points of cluster structure in a temporal network. Peixoto and Rosvall [27] proposed a method to simultaneously detect the clusters and timescales in temporal network, however, they model the temporal network as a single sequence of tokens (similar to [24]) that represent temporal edges, and their timescale inference refers to the number of tokens in the memory of a Markov chain that models such a sequence. In our view, these works focus on mesoscale structures, and take a coarse grained view of temporal networks, while in this work, we propose a complementary approach by focusing on local correlations between temporal edges incident on a node and subsequent temporal edges emanating from it. Among the works that took a fine-grained view, Williams et al. [7] investigated correlations between the temporal edges, however, they, too, considered sequences of edges that do not have any nodes in common, and therefore are not directly related to time-respecting paths. Scholtes et al. [16] found that correlations between edges on time-respecting paths affect centralities, and modeled the time-respecting paths with higher-order models, and found that this approach improves the centrality rankings, and identified the issue of timescale detection, which our work complements. Our work also complements Pfitzner et al. [28] which introduces betweenness preference that can be used to study over- and under-represented time-respected paths in temporal networks, but does not address the problem of detecting the timescales at which these paths occur. To the best of our knowledge, our work is the first to address the problem of causal timescale detection in temporal networks.

---

[1]We note that the processes on and of the temporal network may interact [20], and thus blur the distinction.

**The causal path entropy.** We address the issue of timescale detection by analysing time-respecting paths $\mathcal{P}_{\Delta t}$ of length $k$ for different timescales $\Delta t = [\delta_{\min}, \delta_{\max}]$. Assuming casual interactions where paths incident on a given vertex effect the subsequent edges emanating from it, such time-respecting paths correspond to potential causal influences. In the context of timescale detection we are interested in the timescales where such paths become most predictable given their first $k-1$ steps. To quantify this dependence we propose the "causal path entropy" which, given the set of all of time-respecting paths $\mathcal{P}_{\Delta t}$ of length $k$ and timescale $\Delta t$, is defined as the conditional entropy of the last node $v_k$ given the sub-path $(v_0, v_1, \ldots, v_{k-1})$:

$$\mathcal{H}(\mathcal{P}_{\Delta t}) = H(v_k | v_0, \ldots, v_{k-1}) = H(v_0, \ldots, v_k) - H(v_0, \ldots, v_{k-1}), \tag{2}$$

where $H(P) = -\sum_i p_i \log(p_i)$ is the Shannon entropy. The identity in Eq. 2 can be obtained by applying the chain rule (see Appendix for derivation). By definition $\mathcal{H}(\mathcal{P}_{\Delta t})$ measures the average uncertainty in the last step of time-respecting paths given the $k-1$ previous steps. As the causal path entropy measures the strength of time ordered correlations it can be also related to the concept of Granger causality [29]. A lower value of the entropy indicates a high correlation between the memory of time-respecting paths and subsequent steps. Hence the $\Delta t$ for which the entropy reaches its minimum gives us the timescale for which causal paths become most predictable, i.e. where the correlations between subsequent temporal edges are the most pronounced. The entropy can also be defined for a single node $v$, by simply fixing $v_{k-1} = v$, allowing for a more fine grained analysis that could be important if nodes differ significantly with respect to the timescales they operate on. Given a timescale $\Delta t$, the entropy can be estimated using the counts of time-respecting paths of length $k$ e.g. using the methods from [30, 31].

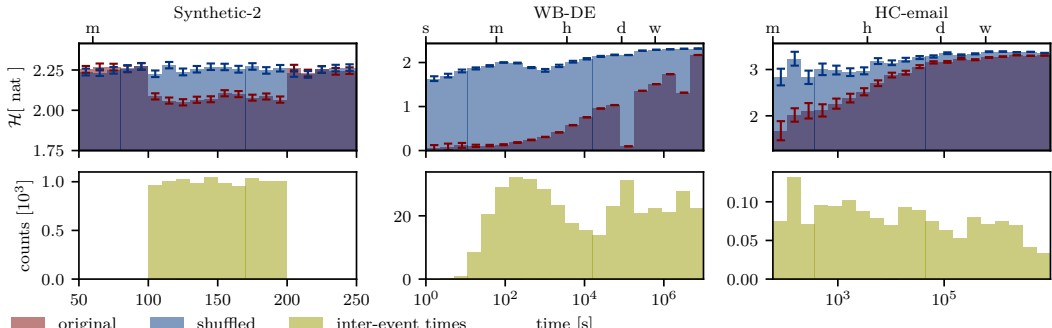

**Figure 1:** Top: causal path entropy as a function of causal temporal scales in datasets Synthetic-2, WB-DE, and HC-email (transparent red) and in the temporal networks with shuffled timestamps (transparent blue). The height of a bar represents causal path entropy (error bars represent the estimation error) and the $x$-limits of a bar represent the interval $\Delta t = [\delta_{\min}, \delta_{\max}]$ on which the causal path entropy was measured. We indicate on $x$-axis the timescales of one minute (m), hour (h), day (d), week (w), and year (y). We observe that the causal path entropy differs more between the original and the shuffled network at causal timescales. Bottom: histogram of causal inter-event times.

In practice the causal path entropy is estimated from path counts by assuming multinomial distributions with respective probabilities $p(v_0, \ldots, v_{k-1})$ and $p(v_0, \ldots, v_k)$. The estimation of the entropy can be challenging especially for small ranges of timescales, since the temporal network can get temporally disconnected resulting in very few paths of order $k$ being observed. As a result we require an efficient method for estimating the entropy that performs well even in such under-sampled regimes. The simplest estimator of a multinomial distribution, called the plug-in estimator, is based on the maximum likelihood estimation which however is known to severely underestimate the entropy in the undersampled regime and has various corrections (e.g. [32, 33]). An alternative to the plug-in estimator is to follow a Bayesian approach which results in entropy estimators that strongly depend on the choice of prior. To counteract this dependency the NSB estimator [34] directly infers the entropy from the counts by averaging over different priors for the transition probabilities, rather than inferring transition probabilities. Being a Bayesian method, the NSB estimator can also be used to quantify the uncertainty of the estimate. More specifically, assuming that the estimates of $H(v_0, \ldots, v_k)$ and $H(v_0, \ldots, v_{k-1})$ have independent errors $\sigma_k$ and $\sigma_{k-1}$, we can approximate the total error of the estimate as $\sigma = (\sigma_k^2 + \sigma_{k-1}^2)^{1/2}$. As the NSB estimator requires the size of the alphabet to be known it is most suitable for cases where the number of nodes is fixed and improves further if the set of

edges that can occur are known a priori as this further restricts the number of potential paths. In cases when the number of nodes in the system is unknown, the Pitman-Yor Mixture entropy estimator [35] could be used instead. The details of our implementation and the computational complexity can be found in the Appendix.

**Experiments.** We first perform synthetic experiments in order to observe the behavior of the causal path entropy in a controlled setting. We simulate temporal network Synthetic-2 with ground truth timescale $\bar{\Delta}t = [100, 200]$ using the procedure described in the Appendix. We also generate a shuffled temporal network, by randomly shuffling edge timestamps, thus destroying the correlations between topological and temporal patterns (while preserving the distribution of edges and the distribution of timestamps). The values of the causal path entropy for the synthetic network and the shuffled network along with the histogram of inter-event times between causal paths is shown in Fig.1. More synthetic examples including results for networks with higher order correlations and multiple timescales found in the Appendix. In the synthetic data sets we find that the causal path entropy behaves as expected and is able to fully recover the timescales of the planted casual interactions from the data. Moreover, this pattern disappears when the timestamps of edges are shuffled, demonstrating that measure correctly captures the dependencies between temporal and topological patterns.

In general, testing in real world data is more challenging due to the lack of a ground truth timescale. In order to circumvent this problem, we consider two types of temporal networks for which we were able to extract ground truth causal paths. First, we consider the public data set of Hillary Clinton's emails [36], for which we extract causal paths in the form of email chains with common subject headers i.e. emails that are forwards or replies to each other. Second, we consider Wikibooks co-editing data sets [37, 38] which contain the editor, the edited article and the timestamp of each edit. We preprocess this data to obtain a temporal network between editors: by placing an edge $(v, w, t)$ whenever editor $w$ edits a given article at time $t$ after it's last editor $v$. We then define causal inter-event times based on the time intervals between successive edits of each article in the data set. More details on these data sets along with results for additional Wikibooks data sets can be found in the Appendix. In both the HC-email and WB-DE data sets we observe that the deviation of the causal path entropy from the baseline closely mirrors the frequency of causal paths and that decreases in the causal path entropy coincide with peaks in the number of causal paths. Moreover, the HC-emails data set, which is an ego-network, demonstrates that the causal path entropy is able to identify causal timescales for individual nodes and that it can be used in cases where the networks is only partially known. We also consider additional temporal networks, namely the email data sets [39, 40] and the SocioPatterns datasets [41–46], for which were unable to obtain ground truth causal paths in the Appendix. Nevertheless for these networks we obtain similar significant timescales for networks of the same type that in accordance with our expectations e.g. typical response times in emails.

We identify three limitations of our approach. First, being based on directed paths the current method is restricted in the types of causal interactions it considers, namely interactions where an incoming link into vertex effects the subsequent links emanating from the vertex. The method could potentially be generalized to other types of casual interactions by considering other temporal patterns. The second limitation of the method is that it can not detect timescales at which the incoming edges to a node change the overall activity of the node without changing the relative frequencies of the outgoing edges. Detecting timescales of such causal influences is thus an open problem. Third, real data can contain time-varying timescales, e.g. during day or night, which might be addressed using time warping techniques.

**Conclusion.** To summarize, the analysis of temporal networks heavily depends on the analysis of time-respecting paths [8, 9, 13, 16, 18, 30]. However, in order to model and analyze the time-respecting paths, we first need to identify the correct timescale. In this work we address this problem by introducing an information theoretic measure, the causal path entropy, that is able to capture timescales at which causal influences occur in temporal networks. Using real world data we demonstrated that the measure can be applied to temporal networks as a whole as well a single nodes and showed that the causal path entropy accurately captures the causal timescales in both synthetic and empirical temporal networks. We further support our findings by observing that the decreases in the causal path entropy coincide with increases in the number of causal paths. The causal path entropy allows system relevant timescales to be inferred from the temporal networks themselves which is crucial for the analysis of temporal networks where inherent timescales are unavailable and hard to measure.

## Acknowledgments

The authors would like to thank Christopher Blöcker, Chester Tan, and Franziska Heeg for valuable comments on the manuscript. LP and IS acknowledge support by the Swiss National Science Foundation, grant 176938.

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

## A    Datasets

In this work we considered synthetic and empirical temporal networks. The details of each temporal network are in Table 1.

| dataset | $|V|$ | $|E|$ | $|\mathcal{E}|$ | $T_{\text{total}}\,[s]$ |
|---------|------|-------|--------|-----------|
| DNC-16 | 1891 | 5598 | 39264 | 8.49e+07 |
| EU-email-1 | 309 | 3031 | 61046 | 6.94e+07 |
| EU-email-2 | 162 | 1772 | 46772 | 6.94e+07 |
| EU-email-3 | 89 | 1506 | 12216 | 6.93e+07 |
| EU-email-4 | 142 | 1375 | 48141 | 6.94e+07 |
| EU-email-A | 986 | 24929 | 332334 | 6.95e+07 |
| Gallery | 10972 | 89034 | 831824 | 6.95e+06 |
| HC-email | 326 | 385 | 8313 | 1.19e+08 |
| Hospital | 75 | 2278 | 64848 | 3.48e+05 |
| Hypertext | 113 | 4392 | 41636 | 2.12e+05 |
| Primary | 242 | 16634 | 251546 | 1.17e+05 |
| School-13 | 327 | 11636 | 377016 | 3.64e+05 |
| Synthetic-1 | 50 | 500 | 30000 | 1e+05 |
| Synthetic-2 | 50 | 500 | 40000 | 1e+05 |
| Synthetic-3 | 50 | 500 | 60000 | 1e+05 |
| Synthetic-4 | 50 | 898 | 40000 | 1e+05 |
| Synthetic-5 | 50 | 500 | 50000 | 1e+05 |
| WB-AR | 1124 | 3334 | 27166 | 3.89e+08 |
| WB-DE | 10999 | 54700 | 464089 | 4.87e+08 |
| WB-FR | 9735 | 53606 | 362094 | 4.88e+08 |
| Work-13 | 92 | 1510 | 19654 | 9.88e+05 |

**Table 1:** The details about temporal networks that we analyzed in the experiments.

To generate synthetic temporal networks Synthetic-1, Synthetic-2 and Synthetic-3 with a ground truth timescale $\bar{\Delta}t = \left[\bar{\delta}_{\min}, \bar{\delta}_{\max}\right]$, we start from a static Erdős-Rényi random graph with $50$ nodes and $500$ directed edges. We sample a random subset $\mathcal{P}_{\text{causal}}$ of $n_{\text{u.p.}} = 500$ unique paths of length $k = 2$ in the static network which correspond to causal influences in the system. We sample with repetition $n_{\text{p}} = 5000$ paths from $\mathcal{P}_{\text{causal}}$ to generate dataset Synthetic-1, $n_{\text{p}} = 10000$ paths to generate dataset Synthetic-2 and $n_{\text{p}} = 20000$ paths to generate dataset Synthetic-3. To add each path $(v_0, v_1, v_2)$ to the temporal network, we sample a random starting time $t$ uniformly from $\left[0, T_{\text{total}} - \bar{\delta}_{\max}\right]$ and create a temporal edge $(v_0, v_1, t)$; we then sample temporal distance $\delta$ between edges on the path (inter-event time) uniformly from $\bar{\Delta}t$ and create the temporal edge $(v_1, v_2, t+\delta)$. We choose $\bar{\Delta}t$ with $\bar{\delta}_{\min} = 100$ and $\bar{\delta}_{\max} = 200$. To add some noise to the system, we uniformly sample $n_{\text{r. e.}} = 20000$ edges from the static graph, and sample their timestamps uniformly from $[0, T_{\text{total}}]$. The temporal network Synthetic-4 contains two timescales relevant for the dynamics. To do so, we generated two different temporal networks based on two random graphs of $50$ nodes (with the same node names) and $500$ edges and based on the different timescales $\Delta t^1 = [50, 100]$ and $\Delta t^2 = [150, 200]$. We used the same procedure as above with parameters $n_{\text{u.p.}} = 500$; $n_{\text{p}} = 5000$; $T_{\text{total}} = 10^5$; $n_{\text{r.e.}} = 10000$. We merged the two temporal networks into one; the details of the resulting network are in Table 1. The dataset Synthetic-5 contains paths of length three. Again, there are $50$ nodes and $500$ edges in the static Erdős Rényi graph. We sample $n_{\text{u.p.}} = 20$ unique paths, we sample $n_{\text{p}} = 20000$ of them, and spread them across $T_{\text{total}} = 10^5$ using the same procedure and timescale $\Delta t = [100, 200]$. We add $n_{\text{r.e}} = 10000$ random edges to the network as noise.

We also use empirical dataset where can get access to the ground truth causal path structure. We consider the bipartite temporal network of Wikibooks co-edits in Arabic (WB-AR), French (WB-FR) and German (WB-DE) [37, 38]. This data contains information about edits on the Wikibooks website: for each edit, we know the editor, the article that was edited, and the time at which the edit occurred. We preprocess this data to obtain a temporal network of editors: if editor $v$ edited an article prior to editor $w$ who edited the same article at time $t$, we assume that a link $(v, w, t)$ occurred in the temporal network of editors. We define causal inter-event times based on the articles: we extract the time intervals between successive edits of each article. In these data, we analyse the timescales of the whole temporal network. Another dataset where we can get access to the ground truth causal

structure is the public data set of Hillary Clinton's emails (HC-email) [36], which contains the sender, the receiver, the timestamp, and the subject of each email. In this data set we analyse the timescales of node representing Hillary Clinton. While sender, receiver and the timestamp form a temporal network, email subjects allow us to obtain causal inter-event times: for each incoming email, we extract the time duration until an email with the same subject was sent. We use the inter-event times between emails with the same subject and the inter-event times of articles for evaluation; the temporal networks contain only the temporal edges and not any additional information about the ground truth timescales.

Finally, we also use empirical temporal networks where we do not know the ground truth causal path structure. Dataset DNC-16 [40] contains emails of the US Democratic National Committee leaked in 2016. Datasets EU-email-1, EU-email-2, EU-email-3, EU-email-4, and EU-email-A [39] contain email correspondence between researchers of an EU institution from first, second, third, fourth and all deparments, repsectively. Datasets Gallery [44], Hospital [42], Hypertext [44], Primary [45, 46], Work-13 [41] and School-13 [43] contain human face-to-face interactions in different settings measured by the SocioPatterns collaborations.

## B    Implementation

Our implementation is based on the path counting methods [30, 31] which we use to obtain counts $n_{v_0 v_1 \ldots v_k}$ of paths $(v_0, v_1, \ldots v_k)$ of length $k$, and counts $n_{v_0 v_1 \ldots v_{k-1}} = \sum_{v_k} n_{v_0 v_1 \ldots v_{k-1}}$ for a given timescale. Note that the computational complexity of the method is dominated by the path counting method whose complexity is upper bound by $\mathcal{O}(|\mathcal{E}| \cdot |V| \cdot k^2 \cdot (m\lambda^{k-2} + \lambda^k))$, where $k$ is the length of the paths, $\lambda$ is the algebraic connectivity of the static network, and $m$ is the maximal number of links within a timescale $\Delta t$. Based on these counts we then estimate the entropies $H(v_0, v_1, \ldots v_k)$ and $H(v_0, v_1 \ldots v_{k-1})$ along with their respective errors using the NSB estimator [34]. Finally, by repeating this procedure over a range of different timescales we identify the timescales for which the entropy is minimized. The code for computing the causal path entropycan be found at [47].

## C    Conditional entropy: The chain rule

For discrete random variables $X$ and $Y$, the definition of the entropy (in nats) is

$$H(X) = -\sum_x p(X = x) \ln p(X = x)$$

and the definition of conditional entropy (in nats) $H(Y|X)$ is:

$$H(Y|X) = -\sum_{x,y} p(X = x, Y = y) \ln \frac{p(X = x, Y = y)}{p(X = x)}$$

In the following, we use the above definitions to derive the chain rule of conditional entropy:

$$H(Y|X) = -\sum_{x,y} p(X = x, Y = y) \left( \ln p(X = x, Y = y) - \ln p(X = x) \right) =$$

$$= -\sum_{x,y} p(X = x, Y = y) \ln p(X = x, Y = y)$$

$$- \left[ -\sum_{x,y} p(X = x, Y = y) \ln(p(X = x))) \right] =$$

$$= H(X, Y) - \left[ -\sum_{x,y} p(Y = y | X = x) p(X = x) \ln(p(X = x))) \right] =$$

$$= H(X, Y) - \left[ -\sum_x p(X = x) \ln(p(X = x))) \left( \sum_y p(Y = y | X = x) \right)^{1} \right] =$$

$$= H(X, Y) - H(X).$$

## D Entropy estimation

In this experiment we test four estimates of the entropy of a multinomial distribution: MLE, Miller [32], Grassberger, and NSB. We vary the sample size and measure the errors of the estimates. For each sample size we repeat the procedure n_repetitions (100) times. First, we generate a random 50-dimensional $\vec{p}$ from a Dirichlet distribution with all concentration parameters $\alpha$ (in code: gen_alpha) $\vec{p} \sim \mathrm{Dir}(\alpha \vec{1})$. The sampled vector $\vec{p}$ represents the ground truth probability distribution, and determines the ground truth entropy $H = \sum_i p_i \ln p_i$. Then, we use the same $\vec{p}$ and generate a random multinomial sample. Using the sample, we infer the entropy rate $\hat{H}$ using four methods: MLE, Miller, Grassberger and NSB. We note the differences between the estimate and the ground truth value, and plot the average errors in those 100 repetitions. Error bars represent intervals between 5th and 95th quantile of the distribution.

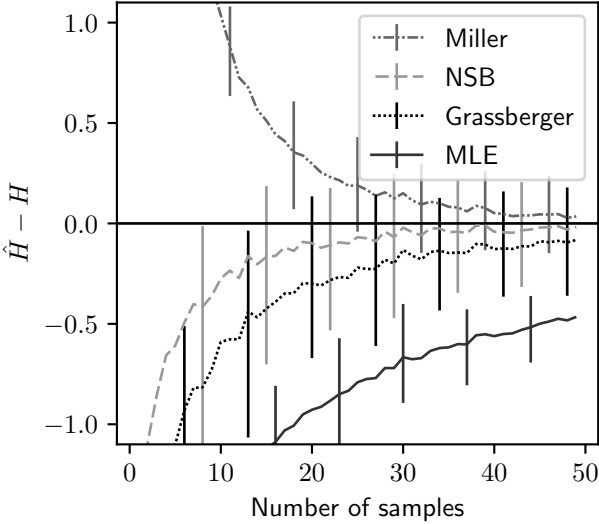

**Figure 2:** E ntropy estimation error as a function of the number of samples. MLE underestimates the entropy for small samples. Grassberger estimates always estimates entropy as $\approx 1.07$ when there is only one sample, and as the data size grows, it approaches the true value. Although NSB estimator was on average a better estimator than the Grassberger estimator, it also had negative bias in our experiments. In contrast, Miller method overestimates the entropy for small data sizes. Error bars denote intervals between 5th and 95th quantile.

# E    Comparison of Entropy Estimates on the Second-Order Transition Matrix

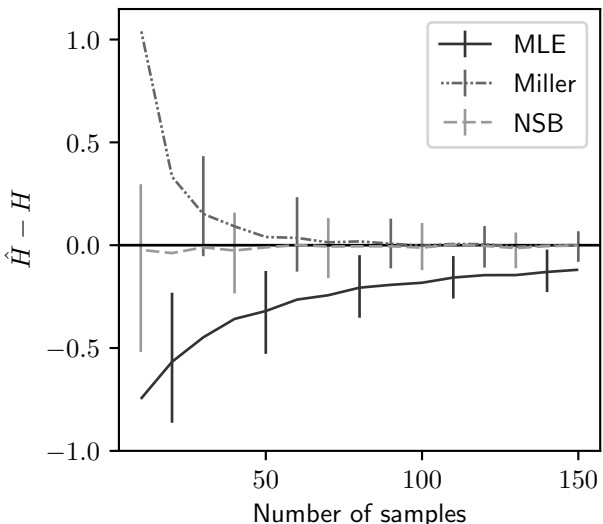

**Figure 3:** We generate random two-hop transition matrix $T_{gt}$ in the form of $p(v_3|v_1, v_2)$, where every row of the matrix represents a different memory $v_1, v_2$ and every column represent different final node $v_3$. We first generate an $\alpha_0$ for that matrix as a draw from a gamma distribution with hyper-parameter 2, then use it to generate every row of the matrix $T_{gt}$ using $\alpha_0$ as the hyperparameters of the Dirichlet distribution. We use the same hyperparameter to generate a probability distribution of $p(v_1, v_2)$. Given the transition matrix $T_{gt}$ and a random probability distribution of memory $v_1, v_2$, we generate a matrix of probabilities $p(v_1, v_2, v_3)$ and from it a matrix of counts $c(v_3|v_1, v_2)$, such that the total number of counts is equal to the number of samples ($x$-axis). We compute the ground truth entropy $H(v_3|v_1, v_2)$ to the entropies inferred from counts $c$. We present the average difference between the two and the error of this average (bars represent quantiles 5-th and 95-th quantile of the differences). For each number of samples (10, 20, 30, ... 150), we ran 100 trials. The temporal network had 4 nodes, and all 12 edges were possible.

# F    Additional synthetic data

In this section we present results on additional synthetic datasets: Synthetic-1 in Fig. 4, Synthetic-3 in Fig. 5, Synthetic-4 in Fig. 6, and Synthetic-5 in Fig. 7.

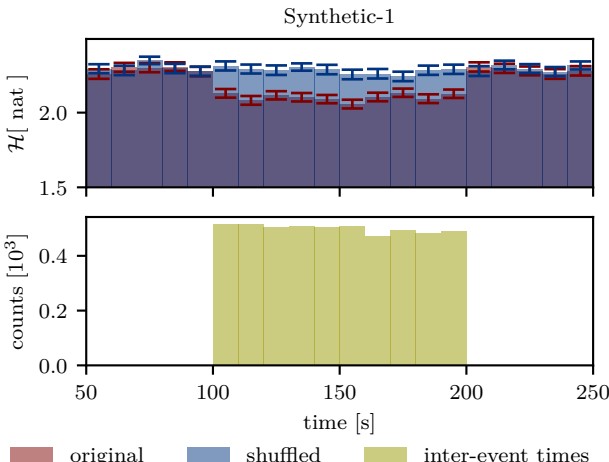

**Figure 4:** Top: causal path entropy as a function of the timescale $\Delta t$ in temporal network Synthetic-1 and in Synthetic-1 with shuffled timestamps. Timescale $\Delta t$ is represented with the $x$-limits of the bar, and causal path entropy is represented as the height of the bar. Error bars indicate the error of the causal path entropy estimates. Bottom: histogram of inter-event times of synthetic causal interactions.

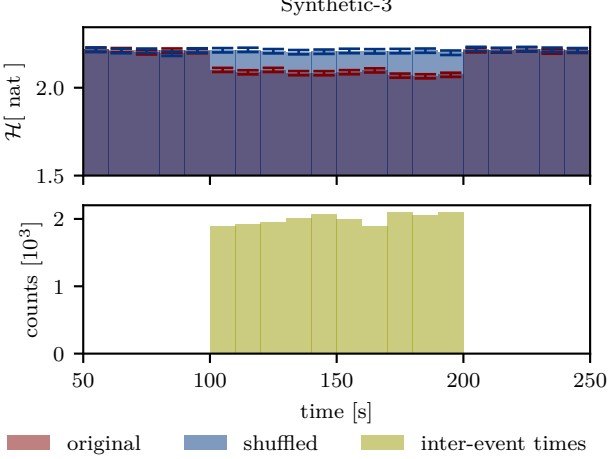

**Figure 5:** Equivalent of Fig. 4, for Synthetic-3.

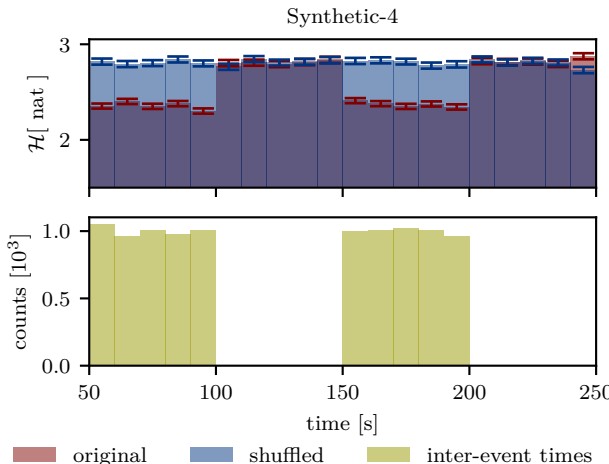

**Figure 6:** Equivalent of Fig. 4, for Synthetic-4 which contains two different timescales.

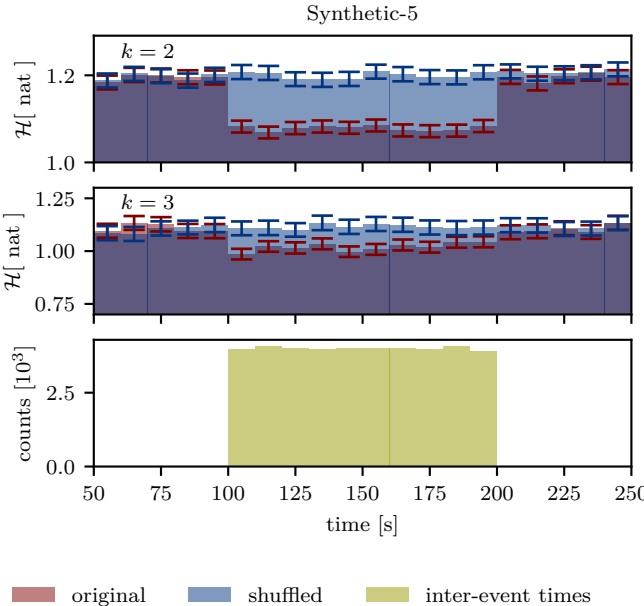

**Figure 7:** Causal path entropy as a function of the timescale $\Delta t$ in temporal network Synthetic-5 and in Synthetic-5 with shuffled timestamps for orders $k = 2$ (top) and $k = 3$ (middle). Timescale $\Delta t$ is represented with the $x$-limits of the bar, and causal path entropy is represented as the height of the bar. Error bars indicate the error of the causal path entropy estimates. Bottom: histogram of inter-event times of synthetic causal interactions.

## G  Other empirical data with ground truth

In this section we show results on other wikibooks datasets (Arabic and French) that we used to test the method.

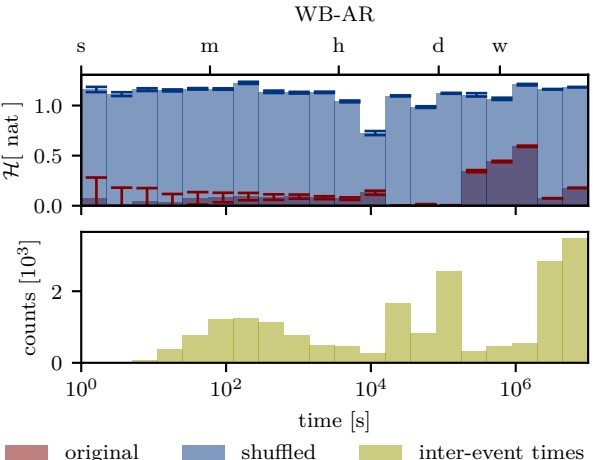

**Figure 8:** Top: causal path entropy as a function of the timescale $\Delta t$ in WB-AR temporal network and of WB-AR temporal network with shuffled timestamps. Timescale $\Delta t$ is represented with the $x$-limits of the bar, and causal path entropy is represented as the height of the bar. Error bars indicate the error of the causal path entropy estimates. Bottom: histogram of inter-event times for all articles of edits of the same article.

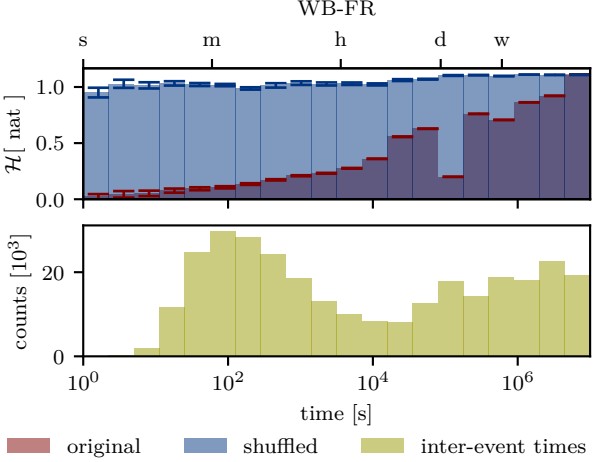

**Figure 9:** Equivalent of Fig. 8 for WB-FR.

## H  Empirical data without the ground truth

In this section, we show multiple datasets in which we do not have access to the ground truth temporal scale, but we observe circadian rhythms and expected patterns of human communications.

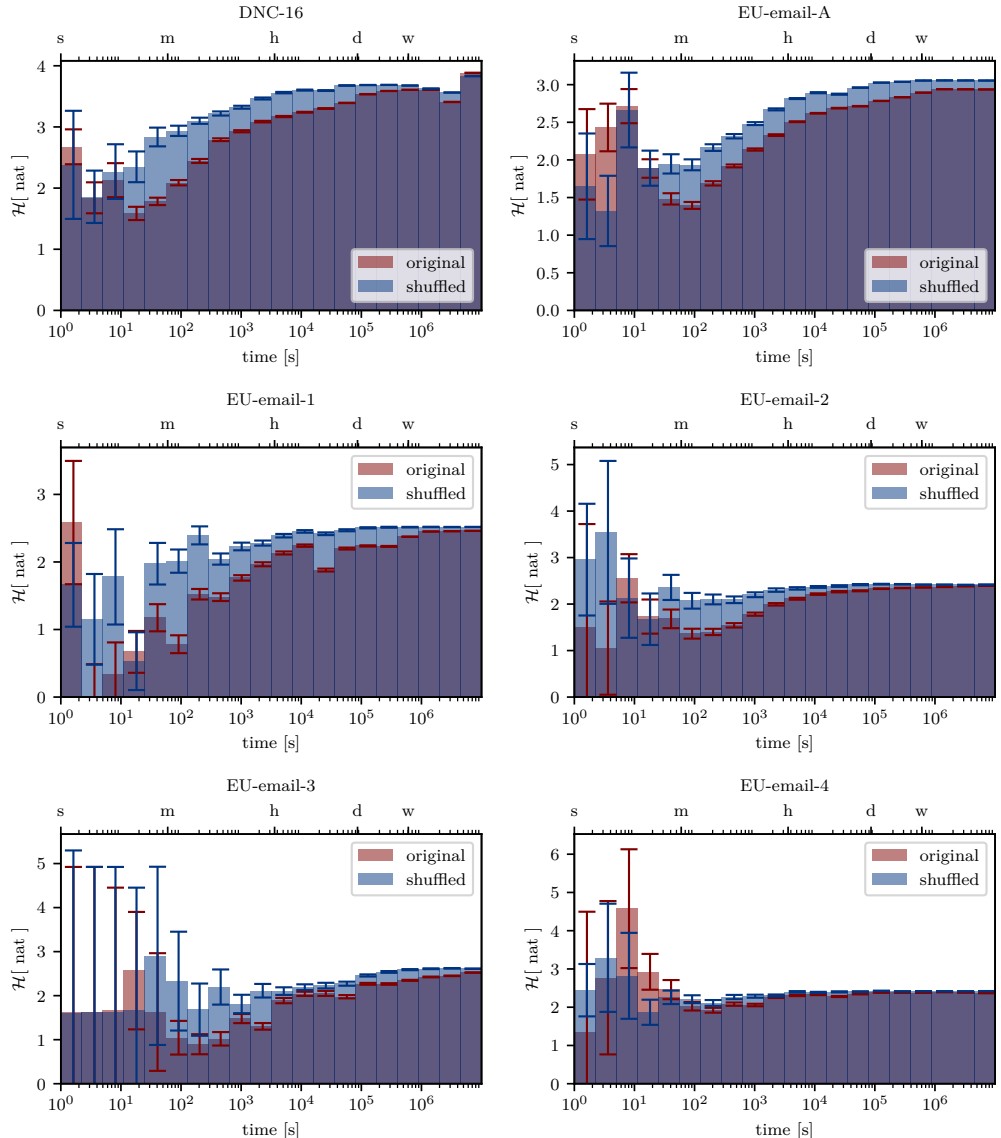

**Figure 10:** Causal path entropy as a function of the timescale $\Delta t$ in temporal networks of email correspondence. For each temporal network, we show the causal path entropy of the original and of a shuffled network. Timescale $\Delta t$ is represented with the $x$-limits of the bar, and causal path entropy is represented as the height of the bar. Error bars indicate the error of the causal path entropy estimates.

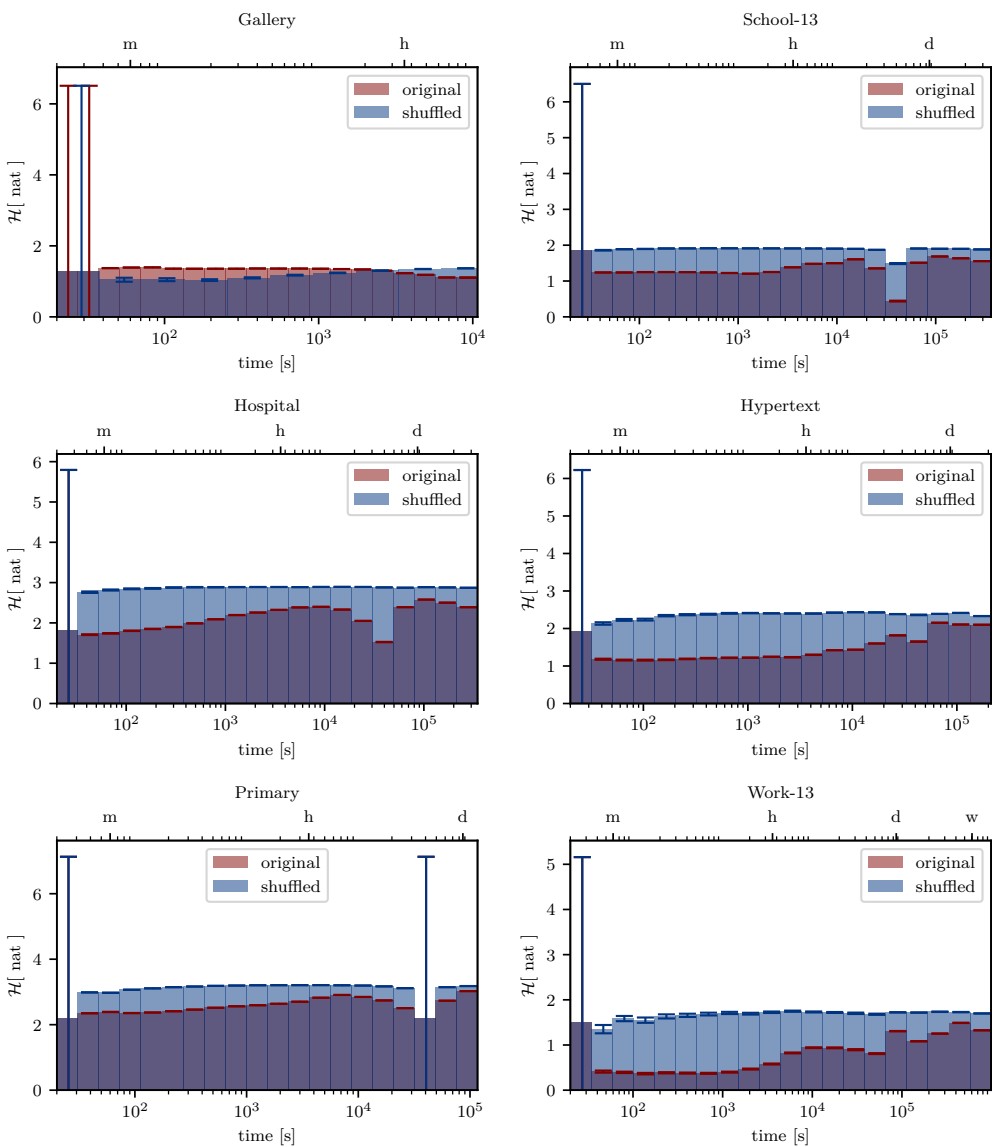

**Figure 11:** Causal path entropy as a function of the timescale $\Delta t$ in temporal networks of human face-to-face interactions measured by the SocioPatterns collaboration. For each temporal network, we show the causal path entropy of the original and of a shuffled network. Timescale $\Delta t$ is represented with the $x$-limits of the bar, and causal path entropy is represented as the height of the bar. Error bars indicate the error of the causal path entropy estimates.

