# OpenReview forum: "Higher-Order Patterns Reveal Causal Temporal Scales in Time Series Network Data"
_logconference.io/LOG/2022/Conference — LoG 2022 Poster_

### Official Review · Reviewer_gDXh · 2022-10-03

**Overall Score:** 8
**Confidence:** 4

**Review:**

The extended abstract proposes an information-theoretic measure for determining the relevant temporal scales in temporal/dynamic networks. The authors show that the resulting method indeed discovers time scales at which paths in the network are created, both for synthetic and real-world datasets.

Discovering the temporal scales in a dynamic graph is a relevant problem and the authors propose a simple, yet effective solution to it. The rationale behind their solution (i.e., that the correct temporal time scale is such that time-respecting paths are most "predictable") is intuitively appealing. The experimental evaluation is valid and appropriate for a extended abstract. The discussion of the related literature is exemplary. Compared to these strengths, I see only three weaknesses, one of which is minor and one of which is probably out of scope for an extended abstract.

1) The manuscript requires further proof-reading (not in terms of language, but in terms of minor typos and consistency of notation; see below).
2) While a measure for choosing temporal scales is proposed and evaluated, what is currently missing are:
2.1) the actual choice, i.e., what time-scales will eventually be used to study the networks? I assume that the answer to this is related to minima in the (derivative of the) evolution of the causal path entropy, but this needs to be stated explicitly.
2.2) how the respective choice compares with methods proposed in the literature (even though they may solve slightly different problems or are designed for slightly different settings); candidates would be your reference [27]; and
2.3) how well do the detected time scales perform on downstream tasks (e.g., clustering, community detection, centrality computation, etc.) where the time scales are required?
3) The rationale that "correct" time scales are characterized by low-entropy time-respecting paths is intuitive. However, one can imagine settings in which time series network data contain periodic patterns that are quite uninteresting; e.g., weekly moderator announcements in newsgroups. Such time scales would then be characterized by particularly low entropy, thus dominating other time scales and potentially "hiding" interesting behavior at longer or shorter time scales. How can this problem be accounted for?

Minor Issues:
- For larger $k$, which may be necessary for studying complex behavior, the sample complexity of the NSB estimator will increase unfavorably. Do the authors have any recommendations?
- In line 106, the authors assume independence between the errors of the estimators for $0,...,k-1$ and for $0,...,k$. How can this independence assumption be justified?

Minor Comments:
- l17-19: This sentence is hard to read.
- l21: aN outgoing message
- l23: takeS place
- l24: CAUSAL
- l72-75: This sentence has many "and"s.
- Consistency: time-scales vs. time scales, plug-in vs. plug in, axis labels in the main part and the appendix of the paper, etc.
- l108: a priorI
- l116: part, WE validate
- l164: remove additional "Fig."

*EDIT* during discussion phase: I lowered my score to a weak accept, concurring with another reviewer's comment that the paper is not clear about how causal time scales can be distinguished from "associative" time scales.

*EDIT* I improved the score again to a clear accept based on the classifications provided by the authors.

---

### Official Review · Reviewer_DvUN · 2022-10-17

**Overall Score:** 6
**Confidence:** 3

**Review:**

This submission considers time series of network data wherein individual, directed edges arrive at successive time points. Given a temporal network, it computes a conditional entropy value associated to individual vertices (i.e., nodes) in paths. Plots and histograms of this entropy value are compared and contrasted across different time window lengths (i.e., temporal resolutions), both for simulated and real datasets.

The temporal analysis of dynamic network-valued data is indeed an active area of ongoing research. I share the authors' interest in this topic. That being said, overall this submission provides only a limited exploration of a few basic concepts. Based on my reading, the submission does not achieve sufficient novelty or substantive content. There are multiple specific issues which further lead me to suggest "reject". Please see below.

The absence of an abstract is a glaring omission.

The authors mention "causal" throughout the paper (and title), in particular in their neologism "causal path entropy". Yet, the meaning of "causal" is never clarified, either formally or informally. This is highly problematic because in the current version, "causal" is merely evocative but not concrete. What connections, if any, are the authors trying make with *causal inference*? Based on my reading, the usage of "causal" is inappropriate and possibly even misleading.

The authors provide an expression for "causal path entropy" in equation (3), but this presentation is incomplete. The authors do not clearly define $H(\cdot)$ evaluated on vertices in a path $v_{0}, v_{1}, \dots, v_{k}$. It is unclear how to interpret or compute the associated quantities when first reading this.

The authors abruptly transition to a discussion of multinomial trials. Why? What are the connections? Up until this point, the data generating mechanisms or assumptions are not stated.

The authors discuss timescales $\Delta_{t}$. Why do the authors use the index $t$? Does $t$ represent "time"? If so, it is unnecessary and possibly a source of confusion, because no mention of $t$ appears in (or is needed in) $\delta_{\min}$ and $\delta_{\max}$ appearing in the definition of $\Delta_{t}$.

The contributions and conclusions provided by the simulations and data analysis are rather limited.

===== Updated below after the posted revisions =====

The revised paper addresses my concerns raised above.

---

### Official Review · Reviewer_MwfM · 2022-10-19

**Overall Score:** 6
**Confidence:** 3

**Review:**

### Contributions
The paper studies the problem of identifying the time scales of causal interactions in temporal graphs with time-stamped edges.
To address this problem, the paper introduces a method to discover the time scale (modeled as an interval of time stamp deltas) at which a measure called "causal path entropy", defined as the entropy of the last node conditional on its preceding time-respecting sub-path, is minimized for time-respecting paths of length 2.
Through experiments on synthetic and real-world data, the paper attempts to demonstrate that the proposed method works well in practice.

### Strong Points and Weak Points
Strong points:
+ Interesting and practically relevant problem on temporal graphs
+ To the extent I can judge from the paper and the Appendix, the approach appears to be theoretically sound for identifying time scales of _correlations_

Weak points:
- Lack of reproducibility (code and data)
- Weak support for inference of _causal_ claims from correlations
- No comparisons with other methods, sensitivity/robustness checks, or correlations with other temporal measures
- No analysis of computational complexity or scalability

### Recommendation
In its current state, I recommend rejecting the paper, primarily due to its lack of reproducibility and the weak justification of its causal claims.
However, I encourage the authors to make their full materials available – first anonymously, and later with DOIs (e.g., assigned via Zenodo) –, and to elaborate on how their method guarantees the link between correlation and causation (see the first question below).
I am willing to reconsider my judgment after inspecting the reproducibility materials and any additional arguments brought forward in support of the link between correlation and causation (alternatively, the authors could tone down the causality claims).

### Questions
- You write (l. 85/86): "A lower value of the entropy indicates a high correlation between the memory of time respecting paths and subsequent steps." So far, so good. But could you further elaborate on how you establish the relationship between correlation and causation in your framework? Currently, the argument appears to be "it seems to work on data where we have causal ground truth", but is there more?
- Is there any way to tell whether the optimal time scales you identify are "real" (i.e., a characteristic of the data-generating process) or due to random fluctuations in the data (i.e., noise)?
- Why do you restrict to k=2? What happens if you allow for larger k, or multiple ks? How do you identify the time scales/deltas to check when searching for the time scale with minimum entropy?
- Is there any prior work introducing methods to detect temporal time scales? If you are the first to do this, it might be worth highlighting, otherwise, you should mention that in the related work section and highlight how your approach differs.
- Is there any method or "naïve approach" you could compare against in your time-scale detection?
- (How) Is the causal path entropy correlated with other (temporal) graph measures?
- (How) Could you detect if causal paths occur at _multiple_ time scales (e.g., some human activity might follow daily, weekly, monthly, and yearly rhythms)?
- Could you comment on the computational complexity of your method?

### Additional Feedback
- You first introduce "'causal path entropy' (of order k)" in l. 81, and later restrict to k=2 in l. 112, apparently also for all of further analyses. The way this is currently written is a bit confusing, as is the definition of a time scale as the interval from a minimum to a maximum difference in time stamps (l. 80, implicitly referencing the notation from ll. 35/36).
- Your short abstract (not included in the paper) states that you introduce "disorder, an information theoretic measure of temporal networks".
The text of the extended abstract speaks of "causal path entropy" only, while "disorder" still features in the labels of some figures.
That is probably not intended, and I would recommend using "causal path entropy" consistently, as it is much more descriptive than the ambiguous "disorder", and still sufficiently short.
- Your title is very Nature/Science/PNAS-like, and it feels a bit over-the-top for this extended abstract. I recommend toning it down, clarifying your contribution already in the title.
- Headings would be helpful for orientation (paragraph-level headings would suffice, and they hardly take space).
- y-axis labels would be appreciated.
- A diagram visualizing your synthetic data generation procedure (e.g., included in the Appendix) would be helpful to understand the procedure you currently describe (only) verbally.
- Make sure to do one more round of language/grammar editing, e.g., to eliminate spelling inconsistencies (esp. w/r/t hyphenation), subject/verb disagreements, missing commas, and typos, both in the main text and in the Appendix.
- In the last column of Table 1, should "y" be "k"?
- "NetSci2022 figures" is probably not the caption you want for Figure 6.
- There is a typo in your running header ("Scalesin" -> "Scales in").

---

### Official Review · Reviewer_49i1 · 2022-10-20

**Overall Score:** 6
**Confidence:** 3

**Review:**

### Summary and recommendation

The paper addresses the problem of estimating the time scale of causal interactions in a temporal network. The proposed approach consists in estimating the time scale by looking at the conditional entropy of the "path" of temporal interactions over the network. The approach seems reasonable and the empirical results suggest that the procedure is able to capture relevant time scales for the interactions; however, a thorough validation would be required to address the extent to which the methodology is effective. Although I have some doubts regarding the technical novelty of this work, I believe that it deserves a spot at the conference as practitioners might find the presented insights and tools useful to build more advanced methods.

As a side note, even if the paper is an extended abstract, the formatting in sections/subsections should be preserved for the sake of readability.

### Comments and issues

Pros:

* The paper is well written and the figures are polished.
* The investigated problem is definitely interesting.
* The proposed solution is simple to implement and could easily be exploited by future works.

Concerns:

1. The motivation for using conditional entropy should be made clearer and the authors should give more intuitions behind this choice.
2. The empirical evaluation is relatively limited, in particular regarding real-world datasets where the discussion of the results should also be expanded. Can you add a comment regarding the specific insights that would be possible to gather from the results on these datasets?
3. The conclusion should also include a short discussion of the current limitations of this work.

Minor:

* Typo in line 164 "Fig Fig 1"

---

### Meta-Review · Area_Chair_r93o · 2022-11-12

**Confidence:** 3
**Recommendation:** Accept

**Meta Review:**

The reviewers have converged on a weak accept rating, and generally seem to agree that the paper considers an interesting and relevant problem, proposes a simple solution method, and that (after some formatting issues were addressed) the paper is readable. The most significant concern raised is the use of the word "causal", which to many readers evokes the association with causal inference. The authors clarified that they have a different meaning in mind, and mention that this usage is established in the temporal networks community. I would like to ask the authors to clearly warn the reader early in the paper not to misinterpret this word, but other than that I do not see this as grounds for rejection. Furthermore, since it does seem intuitively plausible that the method could detect genuine causal relations *under certain assumptions*, it might be worthwhile to investigate the relation to causal inference in future work.

Further concerns that were raised include reproducibility and experimental comparison to baselines. It seems that these concerns were addressed to some degree during the discussion. In my view these concerns are legitimate, but the paper passes the bar for a non-archival short paper.

---

### Decision · Program_Chairs · 2022-11-22

Accept (Poster)